# Investigation and Analysis of Pesticide Residues in Four Common Vegetables and Risk Assessment of Dietary Exposure in Ceramic Capital, China

**DOI:** 10.3390/molecules27196562

**Published:** 2022-10-04

**Authors:** Xingang Meng, Lu Wang, Niao Wang, Luting Chen, Qian Huang

**Affiliations:** 1School of Biological and Environmental Engineering, Jingdezhen University, Jingdezhen 333000, China; 2Liupanshui Ecological Environment Monitoring Center, Liupanshui 553000, China; 3Jingdezhen Nursing School, Jingdezhen 333000, China

**Keywords:** pesticide residues, dietary assessment, vegetables, GC-MS/MS, ceramic capital

## Abstract

In order to understand the basic situation of pesticide residues in vegetables in China’s porcelain capital, four kinds of common vegetables on the market were selected in this study for detection and analysis of pesticide residues. The pesticide residues in vegetables were analyzed through sample selection, optimization of instrument and equipment conditions, and comparison of detection pass rates. The sampling locations were common vegetables purchasing places such as large and medium-sized supermarkets. QuEChERS method was used as the sample pretreatment, and gas chromatography (GC-MS/MS) was used for quantitative analysis. Finally, the exposure risk of pesticides was assessed according to the test results. The results showed that all the pesticides were detected in four kinds of vegetables, but the detected pesticides did not exceed the national standard (GB 2763-2014, China). Moreover, the target risk coefficient (THQ) and risk index (HI) values of four vegetables were less than one, indicating that the combined and toxic effect of pesticide residual mixed contamination was smaller in four vegetables. Therefore, there was no significant harm from people using these vegetables.

## 1. Introduction

Vegetables are one of the necessary food sources in human daily life. Vegetables are rich in vitamins, which are beneficial to human health. Pesticides are generally chemical drugs and biological drugs, which are mainly used to prevent and control pests and regulate plant growth. The cultivation of vegetables is inseparable from pesticides. A pesticide has the characteristics of being convenient, efficient, and having a fast response, and plays an important role in the production of human agriculture, forestry and animal husbandry. According to the statistics, pesticides were found to reduce the crop loss by about 35–45% during the production process, significantly improving the production efficiency [1]. At the same time, due to the ravage of pests, the number of drugs in vegetable farming was increasing, and occasionally spraying a large number of organophosphorus pesticides, leading to serious pesticide residues in vegetables. Eating poisonous vegetables could cause headaches, dizziness, vomiting, difficulty breathing, coma, and even death, so therefore people pay attention to the safety issues of vegetables.

In the past decade, in order to effectively determine the residue of pesticides in vegetables, scholars have conducted a lot of research [2,3,4,5,6]. First of all, there are many studies mainly focused on the optimization of pesticide residue detection methods in vegetables [7,8,9]. The molecularly imprinted solid-phase extraction coupled with high-performance liquid chromatography was established to determine the three organophosphorus pesticides in vegetables. The results showed that the detection limits of dimethoate, isocarbophos, and methyl parathion was 19.78, 8.73 and 17.41 µg/kg, respectively. The relative standard deviation of five replicates of 0.01 mg/L mixed solution was in the range of 1.8–4.2%. The method was also successfully applied in the determination of organophosphorus pesticides (OPs) in cauliflower samples [10]. The solid-phase microextraction–gas chromatography-flame photometry was developed for the determination of eleven OPs residues in vegetables. The detection limit of chlorpyrifos was 0.01–0.14 µg/L and the quantitative limit was 0.03–0.42 µg/L. The highest detection rate of chlorpyrifos was 0.22–1.68 µg/L. The experimental value was still lower than the maximum residue limits (MRL) stipulated in Malaysian food laws and regulations, which can effectively ensure the quality and safety of vegetables [11]. Pesticide residues in vegetables vary in different regions [12]. For example, the concentrations of the residues of selected insecticides (organo-phosphorous and pyrethroid) and fungicides (triazoles and chloronitriles) in fruits and vegetables collected from Xiamen, China were determined by Gas chromatography with electron capture detector (GC-ECD). Among the 1135 samples tested for pesticide residues, Chinese cabbage, beans and mustard greens were the commodities with the most pesticide residues detected, with 17.2%, 18.9% and 17.2% of the samples exceeding MRLs, respectively. The results showed that despite a high occurrence of pesticide residues in fruits and vegetables from this region, it could not be considered a serious public health problem [13]. The concentration of OPs in fresh vegetables from 214 vegetable samples from Changchun, capital of Jilin Province, one of the most important vegetable producing regions, was tested and the potential health risks to residents were estimated. The results showed that 23.4% of the samples contained OPs higher than the MRL, and only 7.9% did not contain OPs. The average THQ were all less than one and the average HI for adults and children were 0.448 and 0.343, respectively. The conclusion was that residents exposed to the average OPs level may not be at risk of health risks [14]. In addition, there were also research and analyses of vegetables in four different quarters [15] and different seasons [16].

A risk assessment has been carried out based on the experimental results of pesticide residues in vegetables [17,18,19,20,21]. In order to investigate the concentration of pesticide residues and potential human health risks, 118 leaf samples which were studied were collected from the northern part of Chile from 2014–2015. The pesticide residue was determined using gas chromatography and high-performance liquid chromatography. The results showed that the maximum estimated daily intake of carbon disulphide (CS_2_), methamidophos, azoxystrobin and cypermethrin was 0.57, 0.07, 0.06 and 0.05 mg/kg, which was higher than its acceptable daily intake. The conclusion was that residents in the northern part of Chile will not face health risks due to eating veneer vegetables. However, the methamidophos content that was detected in the vegetables can be seen as potential chronic health risks [22]. The residual amount of 23 pesticides in southern Nepalese were analyzed. The total amount of pesticide residues (µg/kg) in eggplant, pepper and tomato were 1.71–231, 4.97–507 and 13.1–3465, respectively. The accumulated diet exposure indicated that the organophosphate had higher HI (HI > 83), while the HI of the organochlorine, acaricide and biological pesticide (HI < 1). The pesticide residue concentration in the IPM field was quite low, which indicated that the IPM system can reduce the risk of diet due to contact with pesticides [23]. A sample of 211 vegetables from 10 different vegetable commodities in the Asir region of Saudi Arabia was evaluated using the QuEChERS method according to European regulations. The results showed that none of the lettuce, cauliflower and carrot samples contained pesticide residues, 145 samples contained pesticide residues lower than MRL, 44 samples exceeded MRL, and often exceeded MRL in pepper and cucumber [24]. Data on pesticide residues in vegetables and vegetable consumption were used to assess the exposure of Seoul residents to pesticide residues in vegetables. In the vegetable samples, 86.1% contained no measurable pesticides, and 1.4% contained residues exceeding the MRL. Although some pesticide residues were high, they do not exceed the standard limit, so they may not be considered a serious public health problem [25].

Based on the above-mentioned literature review of vegetable pesticide residues and dietary risk assessment, all countries in the world were extremely concerned about food safety issues, based on this consideration, a study on pesticide residues in vegetables was conducted in Jingdezhen City. Jingdezhen, the porcelain capital of China, is located in the northeast of Jiangxi province, with a subtropical monsoon climate with long light time and heavy rainfall in a year. Due to its special climate conditions, the process of crop production is prone to rampant pests and rapid reproduction, resulting in farmers frequently spraying pesticides in the planting process, as a result, the problem of pesticide residues occurred. The objectives of the present study are as follows: (I) to analyze the causes of the pesticide residues; (II) to evaluate the risk of dietary exposure, and (III) to make a corresponding solution to the vegetables of Jingdezhen City and provide technical support for relevant regulatory authorities.

## 2. Results and Discussion

### 2.1. Method Validation

The pesticide mixed standard solution was prepared and diluted into a standard solution with concentration gradients of different masses, and appropriate GC-M S/MS conditions were set to carry out the determination experiments. When drawing the standard curve, the quality concentration was horizontal coordinates and the peak area was vertical coordinates, so as to obtain a series of linear regression equations (Table 1). In leeks, eight pesticides were determined, the correlation coefficient was greater than 0.99 with a strong correlation between concentration and area for target compound, which also revealed the detection limit (LOD) of 0.3–4.9 µg/kg and the quantitative limit (LOQ) of 1–10 µg/kg; Dichlorvos, omethoate, diazinon, monocrotophos, chlorpyrifos, malathion, methidathion, fenethanil were determined in Chinese cabbage. It can be seen that the LOD was 0.006–4.000 µg/kg and the LOQ was 0.02–13.39 µg/kg. In addition, the linear equation of the six pesticides in cucumbers were consistent with the pesticides corresponding to leeks, and its correlation coefficient was greater than 0.99, which indicated the LOD of 0.2–4.9 µg/kg and the LOQ of 1–10 µg/kg. Lastly, pirimicarb, triazolone, malathion, butachlor, posfolan-methyl, bifenthrin, triazophos, etoxazole were tested in murphy. Except for malathion’s correlation coefficient was less than 0.99, the others were greater than 0.99, which indicated the LOD of 0.1–1.7 µg/kg and the LOQ of 1–5 µg/kg. The test results meet the relevant national standards. Furthermore, this method is fast, sensitive, and accurate, and it is very suitable for the micro detection of pesticides in vegetables.

### 2.2. Results of Pesticide Residues in the Four Vegetables

#### 2.2.1. Leek

The residue of pesticide in the leek can be evaluated through the basic situation of the detection rate of pesticides in the leek and excessive rate related data (See Table 2). The four kinds of pesticides of ethoprophos, chlorpyrifos-methyl, chlorpyrifos, and dichlorvos were not detected, methamidophos, sulfotep, pyridaben, fenethanil were detected. The four pesticides that were detected in the leeks did not exceed the standard and met the pesticide residue standards. The total ion chromatogram in MRM mode, qualitative and quantitative chromatogram of ion pair and mass spectrogram are listed in Appendix A.

#### 2.2.2. Cabbage

The total ion chromatogram in MRM mode, qualitative and quantitative chromatogram of ion pair and mass spectrogram are listed in Appendix A. Table 3 shows that three pesticides of dichlorvos, malathion and methidathion were detected in cabbage, and none of the other five pesticides were detected. None of the three pesticides found in the cabbage exceeded the standard and, therefore, met the pesticide residue standards. This showed that when the vegetables are sprayed with pesticides, the guidelines for the amount of pesticide dosage are obeyed.

#### 2.2.3. Cucumber

Detection results of pesticides residue in cucumber are shown in Table 4. The results indicated that there were four pesticides in cucumber to be detected, respectively, of omethoate, pirimicarb, triazolone, and malathion, while the other four kinds of pesticides were not detected. Overall, the four pesticides were detected in cucumber were lower than the highest limit allowed by the state, meeting the national standards. The total ion chromatogram in MRM mode, qualitative and quantitative chromatogram of ion pair and mass spectrogram are listed in Appendix A.

#### 2.2.4. Potato

Detection results of pesticides residue in potato are shown in Table 5. Only three kinds of pesticides, pirimicarb, bifenthrin and etoxazole, were detected in potatoes, and the remaining five pesticides were not detected. The correlation ratios of the three detected pesticides were also relatively small. Anyway, none of the four pesticides that were detected in potato did not exceed the standard and met the pesticide residue standards. The total ion chromatogram, qualitative and quantitative chromatogram of ion pair and mass spectrogram are listed in Appendix A.

### 2.3. Human Health Risk Assessment of Pesticides in the Four Vegetables

As can be seen from Table 6, the highest average daily pesticide intake (ave EDI) in the leek was sulfotep, the lowest was fenethanil, and their values were 1.494 μg/(kg∙d) and 0.143 μg/(kg∙d), respectively. The maximum daily intake (max EDI) of methamidophos in leek reached 18.945 μg/(kg∙d). The target average risk factor (ave THQ) and target maximum risk factor (max THQ) values in the leeks were less than one, the average risk index (ave HI) and maximum risk index (max HI) values in the leeks were 0.460 and 0.919 μg/(kg∙d). The dichlorvos had the highest average EDI in cabbage which was 4.451 μg/(kg∙d), and the methidathion had the lowest average EDI which was 0.402 μg/(kg∙d). The maximum EDI of dichlorvos in cabbage reached 17.580 μg/(kg∙d). In the small cabbage, the dichlorvos average THQ and maximum THQ were the highest of the three detected pesticides, with values of 0.045 and 0.1758 μg/(kg∙d), the average HI and maximum HI values in the cabbage were 0.069 and 0.221 μg/(kg∙d). In cucumber, the highest average EDI was triazolone, 0.318 μg/(kg∙d), followed by pirimicarb, 0.098 μg/(kg∙d). The highest maximum EDI of the four pesticides detected was triazolone, the value was 0.418 μg/(kg∙d). In cucumber, both average THQ and maximum THQ of malathion were the highest values of the four detected pesticides, and their values were 0.0230 and 0.0578 μg/(kg∙d). In addition, the average HI and maximum HI values in cucumber were 0.040 and 0.082 μg/(kg∙d). Three pesticides were detected in potatoes, namely pirimicarb, bifenthrin, and etoxazole, the highest maximum EDI was pirimicarb, which was 3.158 μg/(kg∙d). The pirimicarb had the highest average EDI which was 1.352 μg/(kg∙d), and the bifenthrin had the lowest average EDI which was 0.316 μg/(kg∙d). In addition, the average HI and maximum HI values in the potatoes were 0.252 and 0.685 μg/(kg∙d).

The results revealed that the obtained maximum value of average THQ and maximum THQ were less than one in the four vegetables. Therefore, it can be known from the Section 3.5.2 Target Hazard Coefficient (THQ) that the four vegetables in Jingdezhen are safe and there is no significant health risk. Furthermore, the average HI and maximum HI values in the four vegetables were less than one, indicating that the combined and toxic effect of pesticide residual mixed contamination were smaller in four vegetables, and can ensure people that they are safe to use. Analysis was obtained from these data, the pesticide residue in the four sample will not have a risk to human health, nor does it produce pesticide residual hybrid contamination toxicity effect.

## 3. Materials and Methods

### 3.1. Chemicals and Materials

The standards of targeted pesticides were purchased from Dr. Ehrenstorfer GmbH (Augsburg, Germany). The mass spectrometric-grade acetic acid, methanol, acetone and acetonitrile were purchased from Merck (Darmstadt, Germany). Analytical-grade HCl, Hac, Anhydrous MgSO_4_, CH_3_COONa, HNO_3_, and H_2_SO_4_ were purchased from Jinshan Chemical Reagent Co. (Chengdu, China). Distilled water was obtained from Watsons Co. Ltd. (Dongguan, China). The syringe filter (nylon, 0.22 μm) was purchased from Peaksharp Technologies (Yibin, China). Primary secondary amine (PSA, 40–60 μm), Octadecyl silane (C18, 40–60 μm), Graphitized carbon black (GCB,40–60 μm) were purchased from Jinshan Chemical Reagent Co. (Chengdu, China).

### 3.2. GC-MS/MS Instrument Conditions

*Chromatographic conditions:* Gas Chromatograph 8890 with Tandem Mass Spectrometer 7000D (Agilent Co., Ltd., Santa Clara, CA, USA) was used to perform all GC–MS/MS analyses. HP-5 MS capillary column (30 m × 0.25 mm × 0.25 μm); inlet temperature: 250 °C; Carrier gas: high-purity helium (He), purity > 99.999%; Column flow rate: 2.0 mL/min; Column temperature program mode: initial temperature 60 °C, hold for 1 min, increase to 150 °C at a rate of 20 °C/min, then increase to 230 °C at 15 °C/min, and then increase to 300 °C at 25 °C/min, keep 5 min; Injection volume: 2 μL; Injection method: pulse split less injection.

*Mass spectrometry conditions:* Detector ionization mode: electron bombardment source (EI); Ion source temperature: 230 °C; Electron energy: 70 eV; Transmission line temperature: 280 °C; Solvent delay time: 5.00 min; Detection mode: multiple ion reaction monitoring mode (MRM).

### 3.3. Preparation of Standard Solutions

Stock standard solutions (100 μg/mL) of the targeted pesticides were prepared separately in methanol. The stock standard solution was diluted with methanol as required. Working standard mixtures tested pesticides were prepared by diluting stock solutions with methanol to a concentration of 0.1–10 μg/mL. A matrix-matched standard calibration was obtained by mixing working standard solutions with blank vegetable extracts. All standard solutions were stored at −18 °C in dark amber bottles until further analysis.

### 3.4. Sample Pretreatment Method

#### 3.4.1. Sample Preparation

The edible part of the vegetables was selected. After shrinking and chopping, the vegetables were fully mixed, and the crushing machine was used to further grind it to make the sample to be tested. Properly store and save it under temperature conditions at −20 °C–16 °C.

#### 3.4.2. Extraction and Purification

According to literature reports, compared the solid phase extraction (SPE), the matrix solid phase dispersion (MSPD), and QuEChERS [26,27] methods, combined with the existing equipment conditions in the laboratory, chose QuEChERS (quick, easy, cheap, effective, rugged, safe) methods which have been developed in recent years. The principle of the method was to extract the sample with an organic solvent, then CH_3_COONa was added and other reagents for the salting layer, dispersed extract, enabling the adsorbent to bind to most of the interference present in the matrix, and purify accordingly. Experimentally selecting the extractant was acetonitrile, commonly used purifying materials have GCB [28], C_18_ [29], PSA [30], etc.

### 3.5. Dietary Exposure Risk Assessment

#### 3.5.1. Acceptable Daily Intake (ADI) and Estimated Daily Intake (EDI)

ADI is the maximum amount of a substance that can be consumed daily in life and will not cause any risk to health. The ADI used in this study was from China National Food Safety Standard GB2763-2021 (the maximum residual limit of pesticides in food safety national standard food).

The daily intake of pesticides depends only in the concentration of pesticides in vegetables, but also depends on the daily intake of vegetables. In addition, human body weight will also affect the tolerance of pesticides. EDI is a concept that considers these factors, the calculation is as follows:(1)EDI=C×Con/Bw,

Formula: EDI—daily pesticide intake [mg∙(kg/d)]; C—Pesticide content in the samples (mg/kg); Con—Citizens consume an average amount of vegetables every day, about 283.8 g/d; Bw—Average adult citizens, about 65 kg.

#### 3.5.2. Target Hazard Quotient (THQ)

THQ takes the ratio of the human pesticide intake and the reference amount of pesticide as the evaluation standard, determines that the human body has the same intake and absorption of the residual pesticide in vegetables, and evaluates the safety of the human body. The method is suitable for a relatively single and simple assessment of pesticide residues.

Target hazard factor calculation formula:(2)THQ=EDI/ADI,

Formula: ADI—the amount of pesticide allowed per day [mg∙(kg/d)].

According to the calculation results, if THQ is less than one, the pesticide residue is relatively safe to human body; otherwise, the health exposure to human body has some risks. The probability of the risk depends on the THQ value, and the greater the value, the greater the corresponding risk.

#### 3.5.3. Hazard Index (HI)

HI was used to evaluate the health risks arising from mixed residual pesticide contamination.
(3)HI=∑THQn,

Based on the calculation results and the size of the value one, if HI less than one, it is determined that multiple pesticide residue mixed pollution in the vegetable will not pose a risk to human health; otherwise, the probability may harm human health, the risk depends on the HI value, the greater the value, the greater the risk generated [31].

## 4. Conclusions

In this study, eight kinds of pesticides in four common vegetables (leek, cabbage, cucumber, potato) were detected and analyzed. QuEChERS technology was used to pretreat vegetable samples, and GC-MS/MS combined detection method was used to extract the pesticides with high detection rate and exceeding the standard. Pesticide residues in the investigated vegetables were analyzed, and the risk of dietary exposure was assessed according to the research results.

The results showed that there are four pesticides in the leeks to be detected, the detection rate of fenethanil was the highest; the highest detection rate in the small cabbage was the malathion; the omethoate have the highest detection rate in the cucumber; in the potato, the bifenthrin detection rate was the highest. From the pesticide residue detection of four kinds of vegetables, it can be found that all the detected pesticides did not exceed the standard, which also indicates that the pesticide dosage of these four kinds of vegetables was strictly controlled, and there was no abuse of pesticides or massive spraying. According to the study, the calculation found that the THQ and HI values of the four vegetables were less than one. As a whole, the pesticide residues in leek, cabbage, cucumber and potato in China’s Porcelain Capital will not cause risks and compound pollution to human health. Nevertheless, it is recommended to investigate the continuous monitoring of pesticide residues in vegetables and stricter supervision.

## Figures and Tables

**Table 1 molecules-27-06562-t001:** The standard curve, LOD, LOQ of different pesticide in the four vegetables.

Vegetable Species	Name of Pesticide	Standard Curve	r^2^	LOD (µg/kg)	LOQ (µg/kg)
Leeks	Methamidophos	y = 100.4x + 641.2	0.9993	3.50	10.00
	Ethoprophos	y = 999.1x + 1617.1	0.9991	2.00	5.00
	Sulfotep	y = 959.5x + 1585.9	0.9950	1.80	10.00
	Chlorpyrifos-methyl	y = 1007.6x + 631.3	0.9998	0.60	1.00
	Chlorpyrifos	y = 1049.4x + 1159.7	0.9979	0.30	1.00
	Dichlorvos	y = 953.4x + 687.8	0.9939	2.20	10.00
	Pyridaben	y = 1167.7x − 471.0	0.9961	0.60	1.00
	Fenethanil	y = 973.2x + 1229.9	0.9926	4.90	10.00
Cabbage	Dichlorvos	y = 1.2925x + 0.7572	0.9949	2.00	6.67
	Omethoate	y = 1.1614x + 1.7412	0.9656	4.00	13.39
	Diazinon	y = 1.0426x + 1.6495	0.9726	0.25	0.84
	Monocrotophos	y = 1.2682x + 0.8314	0.9858	0.25	0.84
	Chlorpyrifos	y = 1.1254x + 1.3404	0.9495	0.25	0.84
	Malathion	y = 0.9084x + 0.9941	0.9938	0.50	1.67
	Methidathion	y = 1.004x + 0.6412	0.9993	0.12	0.41
	Fenethanil	y = 0.9991x + 1.6171	0.9991	0.01	0.02
Cucumber	Omethoate	y = 1007.6x + 631.3	0.9998	0.60	1.00
	Pirimicarb	y = 959.5x + 1585.9	0.9950	1.80	10.00
	Metalaxyl	y = 100.4x + 641.2	0.9993	3.50	10.00
	Triazolone	y = 1007.6x + 631.3	0.9998	0.40	1.00
	Malathion	y = 1049.4x + 1159.7	0.9979	0.20	1.00
	Fenitrothion	y = 953.4x + 687.8	0.9939	2.20	10.00
	Pendimethalin	y = 973.2x + 1229.9	0.9926	4.90	10.00
	Methidathion	y = 1167.7x − 471.0	0.9961	0.60	1.00
Murphy	Pirimicarb	y = 906.9x + 1239.6	0.9956	1.50	5.00
	Triazolone	y = 1292.5x + 757.2	0.9949	1.70	5.00
	Malathion	y = 1161.4x + 174.12	0.9656	1.00	2.00
	Butachlor	y = 1042.6x + 1649.5	0.9926	0.20	1.00
	Posfolan-methyl	y = 1268.2x + 831.4	0.9958	0.30	1.00
	Bifenthrin	y = 1125.4x + 1340.4	0.9995	0.10	1.00
	Triazophos	y = 616.7x + 1543.3	0.9932	0.20	1.00
	Etoxazole	y = 1023.0x + 744.6	0.9972	0.20	1.0

Notes: “r^2^” is the correlation coefficient, “LOD” is the lowest detection limit, “LOQ” indicates the lowest limit of quantification.

**Table 2 molecules-27-06562-t002:** Detection results of pesticides residue in leek.

Order Number	Name of Pesticide	Detection Amount	The Detection Rate (%)	Excessive Scalar	Over Standard Rate (%)
1	Methamidophos	4	20	N.D.	N.D.
2	Ethoprophos	N.D.	N.D.	N.D.	N.D.
3	Sulfotep	2	10	N.D.	N.D.
4	Chlorpyrifos-methyl	N.D.	N.D.	N.D.	N.D.
5	Chlorpyrifos	N.D.	N.D.	N.D.	N.D.
6	Dichlorvos	N.D.	N.D.	N.D.	N.D.
7	Pyridaben	1	5	N.D.	N.D.
8	Fenethanil	7	35	N.D.	N.D.

Notes “N.D.” that means the maximum value of no target pesticide, or the target pesticide did not exceed the national standard in the testing process.

**Table 3 molecules-27-06562-t003:** Detection results of pesticides residue in cabbage.

Order Number	Name of Pesticide	Detection Amount	The Detection Rate (%)	Excessive Scalar	Over Standard Rate (%)
1	Dichlorvos	5	25	N.D.	N.D.
2	Omethoate	N.D.	N.D.	N.D.	N.D.
3	Diazinon	N.D.	N.D.	N.D.	N.D.
4	Monocrotophos	N.D.	N.D.	N.D.	N.D.
5	Chlorpyrifos	N.D.	N.D.	N.D.	N.D.
6	Malathion	8	30.	N.D.	N.D.
7	Methidathion	4	20	N.D.	N.D.
8	Fenethanil	N.D.	N.D.	N.D.	N.D.

Notes: “N.D.” that means the maximum value of no target pesticide, or the target pesticide did not exceed the national standard in the testing process.

**Table 4 molecules-27-06562-t004:** Detection results of pesticides residue in cucumber.

Order Number	Name of Pesticide	Detection Amount	The Detection Rate (%)	Excessive Scalar	Over Standard Rate (%)
1	Omethoate	4	20	N.D.	N.D.
2	Pirimicarb	1	5	N.D.	N.D.
3	Metalaxyl	N.D.	N.D.	N.D.	N.D.
4	Triazolone	2	10	N.D.	N.D.
5	Malathion	3	15	N.D.	N.D.
6	Fenitrothion	N.D.	N.D.	N.D.	N.D.
7	Pendimethalin	N.D.	N.D.	N.D.	N.D.
8	Methidathion	N.D.	N.D.	N.D.	N.D.

Notes: “N.D.” that means the maximum value of no target pesticide, or the target pesticide did not exceed the national standard in the testing process.

**Table 5 molecules-27-06562-t005:** Detection results of pesticides residue in potato.

Order Number	Name of Pesticide	Detection Amount	The Detection Rate (%)	Excessive Scalar	Over Standard Rate (%)
1	Pirimicarb	1	5	N.D.	N.D.
2	Triazolone	N.D.	N.D.	N.D.	N.D.
3	Malathion	N.D.	N.D.	N.D.	N.D.
4	Butachlor	N.D.	N.D.	N.D.	N.D.
5	Posfolan-methyl	N.D.	N.D.	N.D.	N.D.
6	Bifenthrin	6	30	N.D.	N.D.
7	Triazophos	N.D.	N.D.	N.D.	N.D.
8	Etoxazole	2	10	N.D.	N.D.

Notes: “N.D.” that means the maximum value of no target pesticide, or the target pesticide did not exceed the national standard in the testing process.

**Table 6 molecules-27-06562-t006:** Assessment of each pesticide intake and associated health risks among the four vegetables.

Vegetable Species	Name of Pesticide	ADI	Ave EDI (×10^−3^)	Ave THQ	Ave HI	Max EDI (×10^−3^)	Max THQ	Max HI
Leeks	Methamidophos	0.2	0.803	0.0040	0.460	18.945	0.0947	0.919
	Sulfotep	0.08	1.494	0.0187		4.210	0.0526	
	Pyridaben	0.005	0.398	0.0796		1.235	0.2470	
	Fenethanil	0.0004	0.143	0.3575		0.210	0.5250	
Cabbage	Dichlorvos	0.1	4.451	0.045	0.069	17.580	0.1758	0.221
	Malathion	0.07	0.745	0.011		2.085	0.0298	
	Methidathion	0.03	0.402	0.013		0.458	0.0153	
Cucumber	Omethoate	0.3	0.064	0.0002	0.040	0.095	0.0003	0.082
	Pirimicarb	0.08	0.098	0.0012		0.237	0.0030	
	Triazolone	0.02	0.318	0.0159		0.418	0.0209	
	Malathion	0.004	0.092	0.0230		0.231	0.0578	
Murphy	Pirimicarb	0.02	1.352	0.0676	0.252	3.158	0.1579	0.685
	Bifenthrin	0.002	0.316	0.1580		0.832	0.4160	
	Etoxazole	0.02	0.524	0.0262		2.225	0.1113	

Notes: “ADI” is allowable daily intake, “EDI” is estimated daily intake, “THQ” indicates the target hazard quotient, “HI” is Hazard Index, “Ave” is Average, “Max” is Maximum.

## Data Availability

The data presented in this study are available on request from the corresponding author.

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
