# Peer review of "Investigation and Analysis of Pesticide Residues in Four Common Vegetables and Risk Assessment of Dietary Exposure in Ceramic Capital, China"

_molecules, 2022, doi:10.3390/molecules27196562_

Round 1

Reviewer 1 Report

The authors present studies on the situation of pesticide residues in vegetables for four kinds of common vegetables on the Chinese market. The results presented are interesting and valuable. However, the presentation of some of the data,  especially the Figures, should be moved to Supplementary Materials section. The manuscript needs extensive English Style and Grammar corrections and because of that, could be evaluated only after such improvements. Sometimes I do not understand some parts and/or believe the meaning should be different. To help authors, I started from the Abstract and beginning of the manuscript, but the authors have to do all the rest.

See below the text after some corrections have been implemented. For example, the first and last sentences in the Abstract are not fully clear; I do not fully understand what the authors had in mind.

Abstract: In order to understand the basic situation of pesticide residues in vegetables in China's porcelain capital, four kinds of common vegetables on the market were selected in this study. Pesticide residues in vegetables were analyzed by sample selection, optimization of instrument and equipment conditions, and comparison of detection pass rates. The sampling locations were common vegetables purchasing places such as large and medium-sized supermarkets. The QuEChERS method was used as a sample pretreatment, and gas chromatography (GC-MS/MS) was used for quantitative analysis. Finally, the risk of exposure to pesticides was assessed based on the test results. The results showed that all pesticides were detected in four types of vegetables, but the pesticides detected did not exceed the national standard (GB 2763-2014, China). Furthermore, the target risk coefficient (THQ) and risk index (HI) values of four vegetables were less than one, there was no significant harm from people using these vegetables.

Vegetables are one of the necessary food sources in human daily life. Vegetables are rich in vitamins, which are beneficial to human health. Pesticides are generally chemical drugs and biological drugs and are mainly used to prevent and control pests and regulate plant growth. Vegetation cultivation is inseparable from pesticides. The pesticide has the characteristics of convenient, efficient, and fast response and plays an important role in the production of human agriculture, forestry, and animal husbandry. According to incomplete statistics, pesticides were found to reduce crop loss of by about 35% during the production process, significantly improving production efficiency. At the same time, due to the ravage of pests, the number of drugs in vegetable farming was increasing, and occasionally, even spray a large number of organophosphorus pesticides, leading to serious pesticide residues in vegetables. Eating poisonous vegetables can cause headaches, dizziness, vomiting, difficulty breathing, coma, and even death. In recent years, such as poisonous leeks, poisonous bean sprouts, sulfur ginger, etc., there are frequent incidents related to vegetable safety, so people pay attention to the safety issues of vegetables, and are increasingly pursuing ‘Eating’ [1].

Author Response

Reviewer: 1

Comments:

  1. The authors present studies on the situation of pesticide residues in vegetables for four kinds of common vegetables on the Chinese market. The results presented are interesting and valuable. However, the presentation of some of the data, especially the Figures, should be moved to Supplementary Materials section. The manuscript needs extensive English Style and Grammar corrections and because of that, could be evaluated only after such improvements.

Answer: Thank you for your affirmation of our work. I would like to thank you very much for the constructive comments and recommendations on our manuscript. All changes are highlighted in the marked revised manuscript. Thank you for your suggestion. According to your suggestions, We moved Figure 1, Figure 2, Figure 3, Figure 4 into the Supplementary Material section as Figure S1, Figure S2, Figure S3, Figure S4. The paper has been modified by a native English speaker. Other English and related technical changes are indicated in yellow. Thanks again.

  1. Sometimes I do not understand some parts and/or believe the meaning should be different. To help authors, I started from the Abstract and beginning of the manuscript, but the authors have to do all the rest.

See below the text after some corrections have been implemented. For example, the first and last sentences in the Abstract are not fully clear; I do not fully understand what the authors had in mind.

Answer: Thanks for your suggestion. We are so sorry for bringing about troublesome for you because of our careless. In response to this problem, we have clearly explained on lines 12-14 in the revised manuscript, “four kinds of common vegetables on the market were selected in this study” was changed to “four kinds of common vegetables on the market were selected in this study for detection and analysis of pesticide residues”.

And “indicating that the combined and toxic effect of pesticide residual mixed contamination was smaller in four vegetables.” was added in revised manuscript on Line 22-23. The target risk coefficient (THQ) takes the ratio of the human pesticide intake and the reference amount of pesticide as the evaluation standard, determines that the human body has the same intake and absorption of the residual pesticide in vegetables, and evaluates the safety of the human body. Risk index (HI) was used to evaluate the health risks arising from mixed residual pesticide contamination. Based on the calculation results and the size of the value one, if THQ and HI values of four vegetables were less than one, indicating that the combined and toxic effect of pesticide residual mixed contamination was smaller in four vegetables. Thank you for your kind comments.

Other corrections made by the authors and English language improvement. The change was highlighted with yellow background in revised manuscript.

  1. Page 1: In revised manuscript on line 30, are was changed to which are mainly .
  2. Page 1: In revised manuscript on line 32, we have deleted the word "Due to".
  3. Page 1: We have changed " incomplete statistics" to “the statistics” in revised manuscript on line 34. Pesticides were found to reduce the crop loss of by about 35% during the production process, which plays an important role in the production of human agriculture, according to a 1980 article in the American journal Plant Diseases.
  4. Page 1: "35%" was modified to "35%~45%"on Line 34.
  5. Page 1: Reference [1] was moved in from line 40 to line 35.
  6. Page 1, "even" was deleted on Line 37.
  7. Page 1: "In recent years, such as poisonous leeks, poisonous bean sprouts, sulfur ginger, etc., there are frequent incidents related to vegetable safety" was deleted on Line 40 in order to the continuity of the article.
  8. Page 1, “Vegetables are essential to human health, and they can significantly enhance people’s immune system. Some studies have found that vegetables contain a large number of chemicals and minerals that are very important for human health [2]. However, in order to improve vegetable production, many vegetable farmers have applied a large number of pesticides in vegetables to prevent pest breeding, resulting in excessive pesticide residues or illegal pesticides from time to time, which endangers human health. was deleted on Line 41-46 in the previous paper.
  9. Page 1, On line 43 in the previous paper,reference [2] was deleted in order to the continuity of the article. The following references for 3-32 were modified to be 2-31.
  10. Page 1: "many people have studied a lot of research" was changed to "scholars have conducted a lot of research" in revised manuscript on line 43.
  11. Page 1, “many studies mainly focus on the test methods of pesticide residues in vegetables” was modified to “there are many studies mainly focused on the optimization of pesticide residue detection methods in vegetables” on Line 43-45.
  12. Page 2, “were” was modified to “was” on Line 48.
  13. Page 2, “1.8” was modified to “1.8%” on Line 50.
  14. Page 2, “to” was modified to “in” on Line 50.
  15. Page 2:"was established to determine" was modified to "was developed for the determination of" on Line 52-53.
  16. Page 2, “Chen et al. evaluated the residues of selected insecticides (organo-phosphorous and pyrethroid) and fungicides (triazoles and chloronitriles) in fruits and vegetables collected from Xiamen, China. The concentrations of 22 pesticide residues in the recommended pest management index were determined by Gas chromatography with electron capture detector (GC-ECD).” was modified to “the concentrations of the residues of selected insecticides (organo-phosphorous and pyrethroid) and fungicides (triazoles and chloronitriles) in fruits and vegetables collected from Xiamen, China were determined by Gas chromatography with electron capture detector (GC-ECD).” on Line 58-61.
  17. Page 2: "According to the experimental results of pesticide residues in vegetables, many people have carried out risk assessments" was changed to "Risk assessment has been carried out based on the experimental results of pesticide residues in vegetables" in revised manuscript on line 75-76.
  18. Page 2: "118 leaf samples were collected from the northern part of Chile from 2014-2015 were studied" was changed to "118 leaf samples which were studied were collected from the northern part of Chile from 2014-2015" in revised manuscript on line 77-78.
  19. Page 2: "were" was modified to "was" on Line 81.
  20. Page 2: "were detected" was deleted on Line 87.
  21. Page 2, "Samples of 211 vegetables from 10 different vegetable commodities in the Asir region of Saudi Arabia were evaluated. The MRLs for each pesticide in each commodity were evaluated in accordance with European regulations using the QuEChERS method." was modified to "A sample of 211 vegetables from 10 different vegetable commodities in the Asir region of Saudi Arabia was evaluated using the QuEChERS method according to European regulations." on Line 91-93.
  22. Page 3, “they do not exceed the standard limit, so” was added on Line 99-100.
  23. Page 3, We have changed "and this experiment has carried out relevant work investigations in Jingdezhen City" to "based on this consideration, a study on pesticide residues in vegetables was conducted in Jingdezhen City" in revised manuscript on line 103-104.
  24. Page 3, “Jingdezhen, China's porcelain capital, is located in the northeast of Jiangxi, a subtropical monsoon climate, a long light time in a year and large rainfall. Due to its special climate conditions, in the process of crop production, pests are prone to rampant, the rapid growth of pest reproduction and other situations, which prompts farmers to spray pesticides frequently in planting” was modified to “Jingdezhen, the porcelain capital of China, is located in the northeast of Jiangxi province, with a subtropical monsoon climate with long light time and heavy rainfall in a year. Due to its special climate conditions, in the process of crop production, prone to rampant pests, rapid reproduction, resulting in farmers frequently spraying pesticides in the planting process” on Line 104-109.
  25. Page 3, "and then the problem of pesticide residue" was changed to "as a result, the problem of pesticide residues occurred" in revised manuscript on line 108-109.
  26. Page 3, “indicated” was modified to “revealed” on Line 121.
  27. Page 3, “was” was modified to “were” on Line 127.
  28. Page 3, “was” was added on Line 129 and 132.
  29. Page 4, “It can be seen the basic situation of the detection rate of pesticides in the leek and excessive rate related data in the following Table 2 below." was modified to "The residue of pesticide in the leek can be evaluated through the basic situation of the detection rate of pesticides in the leek and excessive rate related data (See Table 2)" in revised manuscript on line 137-138.
  30. Page 4, We have changed " other pesticides were detected." to “methamidophos, sulfotep, pyridaben, fenethanil were detected.” in revised manuscript on line 143.
  31. Page 4, were” was modified to “are” on Line 144.
  32. Page 5, We have changed "was shown" to “shows” in revised manuscript on line 149.
  33. Page 5, “This showed that when the vegetables are sprayed with pesticides, the guidelines for the amount of pesticide dosage are obeyed.” was added on Line 152-154.
  34. Page 5, We have changed "was showed" to “are shown” in revised manuscript on line 159.
  35. Page 5, “other four kinds of pesticides not detected.” was modified to “while the other four kinds of pesticides were not detected.” on Line 165-166.
  36. Page 6, We have changed "was showed" to “are shown” in revised manuscript on line 174.
  37. Page 6, “Therefore” was modified to “Besides” on Line 173.
  38. Page 6, “Table 6” was added in revised manuscript on line 181.
  39. Page 6:"were" was modified to "was" on Line 186, 190, 200, 201 and 205.
  40. Page 6: We have changed "were" to “was” in revised manuscript on line 195.
  41. Page 7, "a correct data" was deleted in revised manuscript on line 206.
  42. Page 7, "So it can be determined according to 1.6.2 Target Hazard Coefficient (THQ) to determine that the four vegetables of Jingdezhen City was safe and there was no significant health risk." was modified to "Therefore, it can be known from the 3.5.2 Target Hazard Coefficient (THQ) that the four vegetables in Jingdezhen are safe and there is no significant health risk." in revised manuscript on line 203-205.
  43. Page 7, “and people can be assured.” was modified to “and can ensure that people are safe to use. on Line 207-208.
  44. Page 7, "in the leek sample" was modified to "in the four sample" in revised manuscript on line 213.
  45. Page 9, "This topic was selected from the four common vegetables, and eight pesticides in the leeks, small cabbage, cucumber and potatoes have conducted pesticide residues" was modified to "In this study, eight kinds of pesticides in four common vegetables (leek, cabbage, cucumber, potato) were detected and analyzed." in revised manuscript on line 294-295.
  46. Page 9, "using GC-MS/MS combined detection method to extract pesticides with high detection rate and exceeding standard, analyzed the residual status of pesticides in the investigated vegetables, and the risk assessment of diet exposure was conducted based on the results of the research" was modified to "and GC-MS/MS combined detection method was used to extract the pesticides with high detection rate and exceeding the standard. Pesticide residues in the investigated vegetables were analyzed, and the risk of dietary exposure was assessed according to the research results." in revised manuscript on line 296-299.
  47. Page 9, "the four pesticides were detected by the cucumber, the omethoate have the highest detection rat" was modified to "the omethoate have the highest detection rate in the cucumber" on line 302.
  48. Page 9, "From the pesticide residue detection of the four vegetables, it can be found that none of the detected pesticides exceed the standard, which can also show that when these four vegetables are planted, there is no abuse of pesticides and pesticide spray when they are strictly controlled by pesticides." was modified to "From the pesticide residue detection of four kinds of vegetables, it can be found that all the detected pesticides did not exceed the standard, which also indicates that the pesticide dosage of these four kinds of vegetables was strictly controlled, and there was no abuse of pesticides or massive spraying." on line 303-306.
  49. Page 9, “Nevertheless, it is recommended to investigate the continuous monitoring of pesticide residues in vegetables and stricter supervision. was added on Line 310-311.
  50. Page 11, “[26] Li, L.H.; Fu, Q.L.; Achal, V.; Liu, Y.L. A comparison of the potential health risk of aluminum and heavy metals in tea leaves and tea infusion of commercially available green tea in Jiangxi, China. Environ Monitor Asses. 2015, 187(5), 1-12.” was deleted.
  51. Page 12, “[31] USEPA. Baseline human health risk assessment. Vasquez boulevard and I-70 superfund site Denver, Co; Denver (Co); 2001” was added on Line 397-398.

Reviewer 2 Report

The article by Xingang Meng et al. is aimed at evaluation of residual amounts of pesticides in vegetables with the use of QuEChERS method for sample pretreatment and GC-MS/MS for quantitative analysis. This is a broad study with extensive experimental work. Despite the fact of focusing on local vegetables the paper contains good level of originality and novelty. Following paper 10.1016/j.snb.2021.130845 should necessarily be referred in this study. Recommendation is accept after minor revision.

Author Response

Reviewer: 2

Recommendation: Publish after minor revisions.

Comments:

The article by Xingang Meng et al. is aimed at evaluation of residual amounts of pesticides in vegetables with the use of QuEChERS method for sample pretreatment and GC-MS/MS for quantitative analysis. This is a broad study with extensive experimental work. Despite the fact of focusing on local vegetables the paper contains good level of originality and novelty. Following paper 10.1016/j.snb.2021.130845 should necessarily be referred in this study. Recommendation is accepted after minor revision.

Answer: Thank you for your affirmation of our work. I would like to thank you very much for the constructive comments and recommendations on our manuscript. The manuscript has carried on the earnest revision with reference to this paper (10.1016/j.snb.2021.130845). The paper has been modified by a native English speaker. All changes are highlighted in the marked revised manuscript. Thanks again for your valuable suggestion.

Other corrections made by the authors and English language improvement. The change was highlighted with yellow background in revised manuscript.

  1. Page 1: In revised manuscript on line 30, are was changed to which are mainly .
  2. Page 1: In revised manuscript on line 32, we have deleted the word "Due to".
  3. Page 1: We have changed " incomplete statistics" to “the statistics” in revised manuscript on line 34. Pesticides were found to reduce the crop loss of by about 35% during the production process, which plays an important role in the production of human agriculture, according to a 1980 article in the American journal Plant Diseases.
  4. Page 1: "35%" was modified to "35%~45%"on Line 34.
  5. Page 1: Reference [1] was moved in from line 40 to line 35.
  6. Page 1, "even" was deleted on Line 37.
  7. Page 1: "In recent years, such as poisonous leeks, poisonous bean sprouts, sulfur ginger, etc., there are frequent incidents related to vegetable safety" was deleted on Line 40 in order to the continuity of the article.
  8. Page 1, “Vegetables are essential to human health, and they can significantly enhance people’s immune system. Some studies have found that vegetables contain a large number of chemicals and minerals that are very important for human health [2]. However, in order to improve vegetable production, many vegetable farmers have applied a large number of pesticides in vegetables to prevent pest breeding, resulting in excessive pesticide residues or illegal pesticides from time to time, which endangers human health. was deleted on Line 41-46 in the previous paper.
  9. Page 1, On line 43 in the previous paper,reference [2] was deleted in order to the continuity of the article. The following references for 3-32 were modified to be 2-31.
  10. Page 1: "many people have studied a lot of research" was changed to "scholars have conducted a lot of research" in revised manuscript on line 43.
  11. Page 1, “many studies mainly focus on the test methods of pesticide residues in vegetables” was modified to “there are many studies mainly focused on the optimization of pesticide residue detection methods in vegetables” on Line 43-45.
  12. Page 2, “were” was modified to “was” on Line 48.
  13. Page 2, “1.8” was modified to “1.8%” on Line 50.
  14. Page 2, “to” was modified to “in” on Line 50.
  15. Page 2:"was established to determine" was modified to "was developed for the determination of" on Line 52-53.
  16. Page 2, “Chen et al. evaluated the residues of selected insecticides (organo-phosphorous and pyrethroid) and fungicides (triazoles and chloronitriles) in fruits and vegetables collected from Xiamen, China. The concentrations of 22 pesticide residues in the recommended pest management index were determined by Gas chromatography with electron capture detector (GC-ECD).” was modified to “the concentrations of the residues of selected insecticides (organo-phosphorous and pyrethroid) and fungicides (triazoles and chloronitriles) in fruits and vegetables collected from Xiamen, China were determined by Gas chromatography with electron capture detector (GC-ECD).” on Line 58-61.
  17. Page 2: "According to the experimental results of pesticide residues in vegetables, many people have carried out risk assessments" was changed to "Risk assessment has been carried out based on the experimental results of pesticide residues in vegetables" in revised manuscript on line 75-76.
  18. Page 2: "118 leaf samples were collected from the northern part of Chile from 2014-2015 were studied" was changed to "118 leaf samples which were studied were collected from the northern part of Chile from 2014-2015" in revised manuscript on line 77-78.
  19. Page 2: "were" was modified to "was" on Line 81.
  20. Page 2: "were detected" was deleted on Line 87.
  21. Page 2, "Samples of 211 vegetables from 10 different vegetable commodities in the Asir region of Saudi Arabia were evaluated. The MRLs for each pesticide in each commodity were evaluated in accordance with European regulations using the QuEChERS method." was modified to "A sample of 211 vegetables from 10 different vegetable commodities in the Asir region of Saudi Arabia was evaluated using the QuEChERS method according to European regulations." on Line 91-93.
  22. Page 3, “they do not exceed the standard limit, so” was added on Line 99-100.
  23. Page 3, We have changed "and this experiment has carried out relevant work investigations in Jingdezhen City" to "based on this consideration, a study on pesticide residues in vegetables was conducted in Jingdezhen City" in revised manuscript on line 103-104.
  24. Page 3, “Jingdezhen, China's porcelain capital, is located in the northeast of Jiangxi, a subtropical monsoon climate, a long light time in a year and large rainfall. Due to its special climate conditions, in the process of crop production, pests are prone to rampant, the rapid growth of pest reproduction and other situations, which prompts farmers to spray pesticides frequently in planting” was modified to “Jingdezhen, the porcelain capital of China, is located in the northeast of Jiangxi province, with a subtropical monsoon climate with long light time and heavy rainfall in a year. Due to its special climate conditions, in the process of crop production, prone to rampant pests, rapid reproduction, resulting in farmers frequently spraying pesticides in the planting process” on Line 104-109.
  25. Page 3, "and then the problem of pesticide residue" was changed to "as a result, the problem of pesticide residues occurred" in revised manuscript on line 108-109.
  26. Page 3, “indicated” was modified to “revealed” on Line 121.
  27. Page 3, “was” was modified to “were” on Line 127.
  28. Page 3, “was” was added on Line 129 and 132.
  29. Page 4, “It can be seen the basic situation of the detection rate of pesticides in the leek and excessive rate related data in the following Table 2 below." was modified to "The residue of pesticide in the leek can be evaluated through the basic situation of the detection rate of pesticides in the leek and excessive rate related data (See Table 2)" in revised manuscript on line 137-138.
  30. Page 4, We have changed " other pesticides were detected." to “methamidophos, sulfotep, pyridaben, fenethanil were detected.” in revised manuscript on line 143.
  31. Page 4, were” was modified to “are” on Line 144.
  32. Page 5, We have changed "was shown" to “shows” in revised manuscript on line 149.
  33. Page 5, “This showed that when the vegetables are sprayed with pesticides, the guidelines for the amount of pesticide dosage are obeyed.” was added on Line 152-154.
  34. Page 5, We have changed "was showed" to “are shown” in revised manuscript on line 159.
  35. Page 5, “other four kinds of pesticides not detected.” was modified to “while the other four kinds of pesticides were not detected.” on Line 165-166.
  36. Page 6, We have changed "was showed" to “are shown” in revised manuscript on line 174.
  37. Page 6, “Therefore” was modified to “Besides” on Line 173.
  38. Page 6, “Table 6” was added in revised manuscript on line 181.
  39. Page 6:"were" was modified to "was" on Line 186, 190, 200, 201 and 205.
  40. Page 6: We have changed "were" to “was” in revised manuscript on line 195.
  41. Page 7, "a correct data" was deleted in revised manuscript on line 206.
  42. Page 7, "So it can be determined according to 1.6.2 Target Hazard Coefficient (THQ) to determine that the four vegetables of Jingdezhen City was safe and there was no significant health risk." was modified to "Therefore, it can be known from the 3.5.2 Target Hazard Coefficient (THQ) that the four vegetables in Jingdezhen are safe and there is no significant health risk." in revised manuscript on line 203-205.
  43. Page 7, “and people can be assured.” was modified to “and can ensure that people are safe to use. on Line 207-208.
  44. Page 7, "in the leek sample" was modified to "in the four sample" in revised manuscript on line 213.
  45. Page 9, "This topic was selected from the four common vegetables, and eight pesticides in the leeks, small cabbage, cucumber and potatoes have conducted pesticide residues" was modified to "In this study, eight kinds of pesticides in four common vegetables (leek, cabbage, cucumber, potato) were detected and analyzed." in revised manuscript on line 294-295.
  46. Page 9, "using GC-MS/MS combined detection method to extract pesticides with high detection rate and exceeding standard, analyzed the residual status of pesticides in the investigated vegetables, and the risk assessment of diet exposure was conducted based on the results of the research" was modified to "and GC-MS/MS combined detection method was used to extract the pesticides with high detection rate and exceeding the standard. Pesticide residues in the investigated vegetables were analyzed, and the risk of dietary exposure was assessed according to the research results." in revised manuscript on line 296-299.
  47. Page 9, "the four pesticides were detected by the cucumber, the omethoate have the highest detection rat" was modified to "the omethoate have the highest detection rate in the cucumber" on line 302.
  48. Page 9, "From the pesticide residue detection of the four vegetables, it can be found that none of the detected pesticides exceed the standard, which can also show that when these four vegetables are planted, there is no abuse of pesticides and pesticide spray when they are strictly controlled by pesticides." was modified to "From the pesticide residue detection of four kinds of vegetables, it can be found that all the detected pesticides did not exceed the standard, which also indicates that the pesticide dosage of these four kinds of vegetables was strictly controlled, and there was no abuse of pesticides or massive spraying." on line 303-306.
  49. Page 9, “Nevertheless, it is recommended to investigate the continuous monitoring of pesticide residues in vegetables and stricter supervision. was added on Line 310-311.
  50. Page 11, “[26] Li, L.H.; Fu, Q.L.; Achal, V.; Liu, Y.L. A comparison of the potential health risk of aluminum and heavy metals in tea leaves and tea infusion of commercially available green tea in Jiangxi, China. Environ Monitor Asses. 2015, 187(5), 1-12.” was deleted.
  51. Page 12, “[31] USEPA. Baseline human health risk assessment. Vasquez boulevard and I-70 superfund site Denver, Co; Denver (Co); 2001” was added on Line 397-398.

Reviewer 3 Report

The present manuscript summarize the results of analysis of  pesticide residues in four common vegetables (& risk assessment of  dietary exposure) in Ceramic Capital, China. The approach of the proposed theme is achieved by simply stating the results obtained. No statistical calculations are made to allow correlations and more complex interpretations of the investigated pesticides concentrations in various vegetables. Moreover, the article presents major deficiencies in terms of the way of expression in English (see attachment) which seriously affects the logical content and the clarity of ideas.  The article cannot be accepted for publication in its current form in Molecules,  a high-class scientific journal.

Author Response

Reviewer: 3

Comments:

The present manuscript summarize the results of analysis of pesticide residues in four common vegetables (& risk assessment of dietary exposure) in Ceramic Capital, China. The approach of the proposed theme is achieved by simply stating the results obtained. No statistical calculations are made to allow correlations and more complex interpretations of the investigated pesticides concentrations in various vegetables. Moreover, the article presents major deficiencies in terms of the way of expression in English (see attachment) which seriously affects the logical content and the clarity of ideas. The article cannot be accepted for publication in its current form in Molecules, a high-class scientific journal.

Answer: Thank you for your affirmation of our work. I would like to thank you very much for the constructive comments and recommendations on our manuscript. Jingdezhen, the porcelain capital of China, is located in the northeast of Jiangxi province, with a subtropical monsoon climate with long light time and heavy rainfall in a year. Due to its special climate conditions, in the process of crop production, prone to rampant pests, rapid reproduction, resulting in farmers frequently spraying pesticides in the planting process, as a result, the problem of pesticide residues occurred. Based on this consideration, it is of great significance to carry out research on pesticide residues in vegetables in Jingdezhen. Our research can provide data support for follow-up related research. The objectives of the present study are as follows: (I) to analyze the causes of the pesticide residues; (II) to evaluate the risk of dietary exposure, and (III) to make a corresponding solution to the vegetables of Jingdezhen City and provide technical support for relevant regulatory authorities. Thank you for your kind comments.

Moreover, in response to this question you raised, “No statistical calculations are made to allow correlations and more complex interpretations of the investigated pesticides concentrations in various vegetables.”, we are very sorry for not in -depth research in this article, and thank you very much for this very valuable opinion. We will consider your valuable suggestion in our next research work. Meanwhile, the manuscript of the English language has carried on the earnest revision. The paper has been modified by a native English speaker. All changes are highlighted in the marked revised manuscript. Thanks again.

Other corrections made by the authors and English language improvement. The change was highlighted with yellow background in revised manuscript.

  1. Page 1: In revised manuscript on line 30, are was changed to which are mainly .
  2. Page 1: In revised manuscript on line 32, we have deleted the word "Due to".
  3. Page 1: We have changed " incomplete statistics" to “the statistics” in revised manuscript on line 34. Pesticides were found to reduce the crop loss of by about 35% during the production process, which plays an important role in the production of human agriculture, according to a 1980 article in the American journal Plant Diseases.
  4. Page 1: "35%" was modified to "35%~45%"on Line 34.
  5. Page 1: Reference [1] was moved in from line 40 to line 35.
  6. Page 1, "even" was deleted on Line 37.
  7. Page 1: "In recent years, such as poisonous leeks, poisonous bean sprouts, sulfur ginger, etc., there are frequent incidents related to vegetable safety" was deleted on Line 40 in order to the continuity of the article.
  8. Page 1, “Vegetables are essential to human health, and they can significantly enhance people’s immune system. Some studies have found that vegetables contain a large number of chemicals and minerals that are very important for human health [2]. However, in order to improve vegetable production, many vegetable farmers have applied a large number of pesticides in vegetables to prevent pest breeding, resulting in excessive pesticide residues or illegal pesticides from time to time, which endangers human health. was deleted on Line 41-46 in the previous paper.
  9. Page 1, On line 43 in the previous paper,reference [2] was deleted in order to the continuity of the article. The following references for 3-32 were modified to be 2-31.
  10. Page 1: "many people have studied a lot of research" was changed to "scholars have conducted a lot of research" in revised manuscript on line 43.
  11. Page 1, “many studies mainly focus on the test methods of pesticide residues in vegetables” was modified to “there are many studies mainly focused on the optimization of pesticide residue detection methods in vegetables” on Line 43-45.
  12. Page 2, “were” was modified to “was” on Line 48.
  13. Page 2, “1.8” was modified to “1.8%” on Line 50.
  14. Page 2, “to” was modified to “in” on Line 50.
  15. Page 2:"was established to determine" was modified to "was developed for the determination of" on Line 52-53.
  16. Page 2, “Chen et al. evaluated the residues of selected insecticides (organo-phosphorous and pyrethroid) and fungicides (triazoles and chloronitriles) in fruits and vegetables collected from Xiamen, China. The concentrations of 22 pesticide residues in the recommended pest management index were determined by Gas chromatography with electron capture detector (GC-ECD).” was modified to “the concentrations of the residues of selected insecticides (organo-phosphorous and pyrethroid) and fungicides (triazoles and chloronitriles) in fruits and vegetables collected from Xiamen, China were determined by Gas chromatography with electron capture detector (GC-ECD).” on Line 58-61.
  17. Page 2: "According to the experimental results of pesticide residues in vegetables, many people have carried out risk assessments" was changed to "Risk assessment has been carried out based on the experimental results of pesticide residues in vegetables" in revised manuscript on line 75-76.
  18. Page 2: "118 leaf samples were collected from the northern part of Chile from 2014-2015 were studied" was changed to "118 leaf samples which were studied were collected from the northern part of Chile from 2014-2015" in revised manuscript on line 77-78.
  19. Page 2: "were" was modified to "was" on Line 81.
  20. Page 2: "were detected" was deleted on Line 87.
  21. Page 2, "Samples of 211 vegetables from 10 different vegetable commodities in the Asir region of Saudi Arabia were evaluated. The MRLs for each pesticide in each commodity were evaluated in accordance with European regulations using the QuEChERS method." was modified to "A sample of 211 vegetables from 10 different vegetable commodities in the Asir region of Saudi Arabia was evaluated using the QuEChERS method according to European regulations." on Line 91-93.
  22. Page 3, “they do not exceed the standard limit, so” was added on Line 99-100.
  23. Page 3, We have changed "and this experiment has carried out relevant work investigations in Jingdezhen City" to "based on this consideration, a study on pesticide residues in vegetables was conducted in Jingdezhen City" in revised manuscript on line 103-104.
  24. Page 3, “Jingdezhen, China's porcelain capital, is located in the northeast of Jiangxi, a subtropical monsoon climate, a long light time in a year and large rainfall. Due to its special climate conditions, in the process of crop production, pests are prone to rampant, the rapid growth of pest reproduction and other situations, which prompts farmers to spray pesticides frequently in planting” was modified to “Jingdezhen, the porcelain capital of China, is located in the northeast of Jiangxi province, with a subtropical monsoon climate with long light time and heavy rainfall in a year. Due to its special climate conditions, in the process of crop production, prone to rampant pests, rapid reproduction, resulting in farmers frequently spraying pesticides in the planting process” on Line 104-109.
  25. Page 3, "and then the problem of pesticide residue" was changed to "as a result, the problem of pesticide residues occurred" in revised manuscript on line 108-109.
  26. Page 3, “indicated” was modified to “revealed” on Line 121.
  27. Page 3, “was” was modified to “were” on Line 127.
  28. Page 3, “was” was added on Line 129 and 132.
  29. Page 4, “It can be seen the basic situation of the detection rate of pesticides in the leek and excessive rate related data in the following Table 2 below." was modified to "The residue of pesticide in the leek can be evaluated through the basic situation of the detection rate of pesticides in the leek and excessive rate related data (See Table 2)" in revised manuscript on line 137-138.
  30. Page 4, We have changed " other pesticides were detected." to “methamidophos, sulfotep, pyridaben, fenethanil were detected.” in revised manuscript on line 143.
  31. Page 4, were” was modified to “are” on Line 144.
  32. Page 5, We have changed "was shown" to “shows” in revised manuscript on line 149.
  33. Page 5, “This showed that when the vegetables are sprayed with pesticides, the guidelines for the amount of pesticide dosage are obeyed.” was added on Line 152-154.
  34. Page 5, We have changed "was showed" to “are shown” in revised manuscript on line 159.
  35. Page 5, “other four kinds of pesticides not detected.” was modified to “while the other four kinds of pesticides were not detected.” on Line 165-166.
  36. Page 6, We have changed "was showed" to “are shown” in revised manuscript on line 174.
  37. Page 6, “Therefore” was modified to “Besides” on Line 173.
  38. Page 6, “Table 6” was added in revised manuscript on line 181.
  39. Page 6:"were" was modified to "was" on Line 186, 190, 200, 201 and 205.
  40. Page 6: We have changed "were" to “was” in revised manuscript on line 195.
  41. Page 7, "a correct data" was deleted in revised manuscript on line 206.
  42. Page 7, "So it can be determined according to 1.6.2 Target Hazard Coefficient (THQ) to determine that the four vegetables of Jingdezhen City was safe and there was no significant health risk." was modified to "Therefore, it can be known from the 3.5.2 Target Hazard Coefficient (THQ) that the four vegetables in Jingdezhen are safe and there is no significant health risk." in revised manuscript on line 203-205.
  43. Page 7, “and people can be assured.” was modified to “and can ensure that people are safe to use. on Line 207-208.
  44. Page 7, "in the leek sample" was modified to "in the four sample" in revised manuscript on line 213.
  45. Page 9, "This topic was selected from the four common vegetables, and eight pesticides in the leeks, small cabbage, cucumber and potatoes have conducted pesticide residues" was modified to "In this study, eight kinds of pesticides in four common vegetables (leek, cabbage, cucumber, potato) were detected and analyzed." in revised manuscript on line 294-295.
  46. Page 9, "using GC-MS/MS combined detection method to extract pesticides with high detection rate and exceeding standard, analyzed the residual status of pesticides in the investigated vegetables, and the risk assessment of diet exposure was conducted based on the results of the research" was modified to "and GC-MS/MS combined detection method was used to extract the pesticides with high detection rate and exceeding the standard. Pesticide residues in the investigated vegetables were analyzed, and the risk of dietary exposure was assessed according to the research results." in revised manuscript on line 296-299.
  47. Page 9, "the four pesticides were detected by the cucumber, the omethoate have the highest detection rat" was modified to "the omethoate have the highest detection rate in the cucumber" on line 302.
  48. Page 9, "From the pesticide residue detection of the four vegetables, it can be found that none of the detected pesticides exceed the standard, which can also show that when these four vegetables are planted, there is no abuse of pesticides and pesticide spray when they are strictly controlled by pesticides." was modified to "From the pesticide residue detection of four kinds of vegetables, it can be found that all the detected pesticides did not exceed the standard, which also indicates that the pesticide dosage of these four kinds of vegetables was strictly controlled, and there was no abuse of pesticides or massive spraying." on line 303-306.
  49. Page 9, “Nevertheless, it is recommended to investigate the continuous monitoring of pesticide residues in vegetables and stricter supervision. was added on Line 310-311.
  50. Page 11, “[26] Li, L.H.; Fu, Q.L.; Achal, V.; Liu, Y.L. A comparison of the potential health risk of aluminum and heavy metals in tea leaves and tea infusion of commercially available green tea in Jiangxi, China. Environ Monitor Asses. 2015, 187(5), 1-12.” was deleted.
  51. Page 12, “[31] USEPA. Baseline human health risk assessment. Vasquez boulevard and I-70 superfund site Denver, Co; Denver (Co); 2001” was added on Line 397-398.

Round 2

Reviewer 1 Report

The authors have intensively improved the manuscript, in the current version the manuscript meets the requirements and can be recommended for publication. 

Reviewer 3 Report

Compared to the original form, the new manuscript has been improved so that it reaches the minimum level required for publication in Molecules.